# Synapse maintenance and restoration in the retina by NGL2

**Florentina Soto[1]\*, Lei Zhao[1], Daniel Kerschensteiner[1,2,3,4]\***

[1]Department of Ophthalmology and Visual Sciences, Washington University School of Medicine, Saint Louis, United States; [2]Department of Neuroscience, Washington University School of Medicine, Saint Louis, United States; [3]Department of Biomedical Engineering, Washington University School of Medicine, Saint Louis, United States; [4]Hope Center for Neurological Disorders, Washington University School of Medicine, Saint Louis, United States

**Abstract** Synaptic cell adhesion molecules (CAMs) promote synapse formation in the developing nervous system. To what extent they maintain and can restore connections in the mature nervous system is unknown. Furthermore, how synaptic CAMs affect the growth of synapse-bearing neurites is unclear. Here, we use adeno-associated viruses (AAVs) to delete, re-, and overexpress the synaptic CAM NGL2 in individual retinal horizontal cells. When we removed NGL2 from horizontal cells, their axons overgrew and formed fewer synapses, irrespective of whether *Ngl2* was deleted during development or in mature circuits. When we re-expressed NGL2 in knockout mice, horizontal cell axon territories and synapse numbers were restored, even if AAVs were injected after phenotypes had developed. Finally, overexpression of NGL2 in wild-type horizontal cells elevated synapse numbers above normal levels. Thus, NGL2 promotes the formation, maintenance, and restoration of synapses in the developing and mature retina, and restricts axon growth throughout life.
DOI: https://doi.org/10.7554/eLife.30388.001

**\*For correspondence:**
sotof@wustl.edu (FS);
kerschensteinerd@wustl.edu (DK)

**Competing interests:** The authors declare that no competing interests exist.

## Introduction

Nervous system functions rely on the precise connectivity of neural circuits. Several synaptic CAMs have been shown to regulate circuit development (*de Wit and Ghosh, 2016*; *Siddiqui and Craig, 2011*). Discrepancies with results obtained in vitro highlight the importance of analyzing the function of synaptic CAMs in vivo (*Südhof, 2017*). Most in vivo studies of synaptic CAMs have compared connectivity of wild-type and germline knockout mice, which remove synaptic CAMs from all cells throughout life. Whether synaptic CAMs act cell-autonomously or not; and whether they maintain synapses and can restore them in mature circuits, therefore, remains unknown. Synapse loss can long precede cell death in neurodegenerative diseases, and the ability to establish connections with mature circuits limits benefits from transplantation of stem-cell-derived neuronal replacements (*Buckingham et al., 2008*; *Gamm et al., 2015*; *Hong et al., 2016*; *Scheff et al., 2006*). Identifying molecular mechanisms that maintain synapses and can promote their restoration is, therefore, an important goal of therapeutic neuroscience.

Netrin-G ligand 2 (NGL2), a synaptic CAM with an extracellular leucine-rich repeat (LRR) domain, regulates synapse development in the hippocampus and the retina (*DeNardo et al., 2012*; *Nishimura-Akiyoshi et al., 2007*; *Soto et al., 2013*). In the hippocampus, NGL2 localizes to the proximal segments of CA1 pyramidal neuron dendrites and promotes the formation of synapses with Schaffer collateral axons (*DeNardo et al., 2012*; *Nishimura-Akiyoshi et al., 2007*). In the retina, NGL2 localizes to the axon tips of horizontal cells and promotes the formation of synapses with rod photoreceptors (*Figure 1—figure supplement 1*) (*Soto et al., 2013*). Whether NGL2 acts cell-autonomously

or not, and whether it maintains synapses and can promote their restoration in mature circuits is unknown.

Here we use AAVs to delete (CRISPR/Cas9), re-, and overexpress NGL2 in individual horizontal cells in the developing and mature retina to address these questions.

## Results

### A strategy for temporally controlled removal of NGL2

We devised the following AAV-mediated CRISPR/Cas9 strategy to delete *Ngl2* with temporal control in individual horizontal cells in vivo. We identified two short guide RNAs (sgRNAs) that reliably introduced frame-shifting insertions and deletions (indels) near the start of the open reading frame of *Ngl2* (see Materials and methods). We generated AAVs (serotype: 1/2) expressing these sgRNAs from a Pol III U6 promoter and tdTomato (tdT) from a Pol II CAG promoter (AAV-sgNGL2-tdT). We injected AAV-sgNGL2-tdT into the vitreous chamber of mice ubiquitously expressing the Cas9 endonuclease (*Platt et al., 2014*) (Cas9 mice, *Figure 1A*). To assess the efficiency of NGL2 removal, we injected AAV-sgNGL2-tdT in newborn (postnatal day 0, P0) Cas9 mice and stained flat-mounted retinas at P30 for NGL2. The NGL2 intensity at axon tips of tdT-positive horizontal cells in Cas9 mice was lower than at neighboring axon tips in 19 of 20 cells (i.e., 95% of cells, *Figure 1B and C*), whereas NGL2 intensity at axon tips of AAV-YFP-infected cells was indistinguishable from neighboring axon tips (*Figure 1C*). At many axon tips of AAV-sgNGL2-tdT-infected cells in Cas9 mice, NGL2 staining was reduced rather than absent. This could be, either because some NGL2 protein remained in horizontal cells expressing sgRNAs, or because multiple horizontal cells contributed to the NGL2 staining at each tip. Given that we injected AAV-sgNGL2-tdT at P0, nearly two weeks before NGL2 is first expressed (*Soto et al., 2013*), residual protein seemed an unlikely explanation. Co-injection of AAVs expressing spectrally separable fluorophores (cyan fluorescent protein [CFP] and tdT) revealed that overlapping horizontal cell axons co-innervate more than 40% of the rods in their shared territory (*Figure 1D and E*). As a population, horizontal cell axons cover the retina

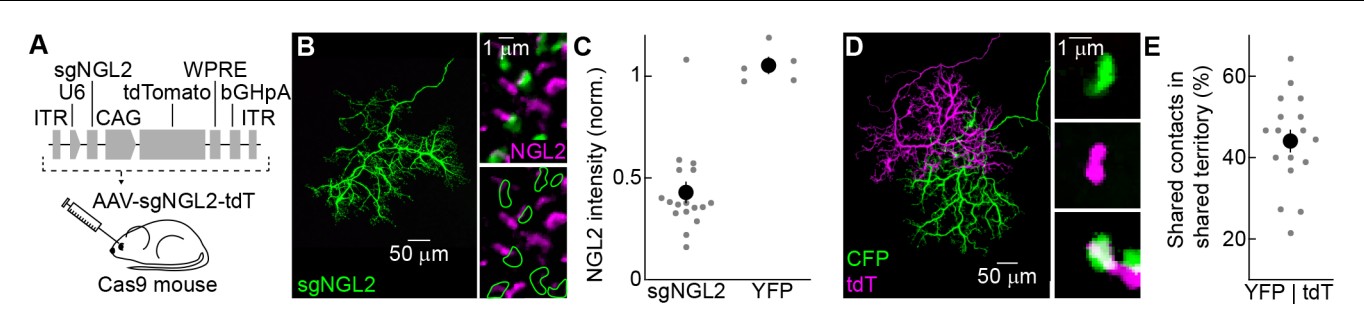

**Figure 1.** AAV-mediated knockout of *Ngl2* in horizontal cells. (A) Schematic illustrating AAV-mediated CRISPR/Cas9 strategy for *Ngl2* knockout in horizontal cells. In AAV-sgNGL2-tdT, small guide RNAs targeting NGL2 (sgNGL2) were expressed from a Pol III U6 promoter, and the red fluorescent protein tdT was expressed from a Pol II CAG promoter. AAV-sgNGL2-tdT was injected intravitreally into Cas9 mice (*Platt et al., 2014*). (B) Representative images of an axon of a horizontal cell infected with AAV-sgNGL2-tdT (injection at P0, analysis at P30) in a Cas9 retina. Left, overview of the axon labeled by tdT; right, magnified excerpts showing NGL2 staining at tips of this axon and overlapping axons of uninfected horizontal cells. (C) Relative NGL2 intensity in axon tips of infected vs. uninfected horizontal cell, for AAV-sgNGL2-tdT (sgNGL2) and AAV-YFP (YFP). Dots show data from single cells compared to its neighbors, the circle (errorbar) indicates the mean (±SEM) of the population. In 19 of 20 horizontal cells (3 mice) infected with AAV-sgNGL2-tdT, the NGL2 intensity was significantly reduced (p<0.01 for each, Wilcoxon rank sum test), whereas NGL2 intensity was unchanged in five of five horizontal cells (2 mice) infected with AAV-YFP. (D) Representative images of two overlapping horizontal cell axons labeled with CFP and tdT, respectively. Left, overview image; right, magnified excerpts from rods contacted by tips of either (top and middle) or both (bottom) axons. (E) Summary data of shared rod contacts (i.e., overlapping axon tips) within the overlapping territory of two horizontal cell axons. Dots show data from individual horizontal cell pairs, the circle (errorbar) indicates the mean (±SEM) of the population.

DOI: https://doi.org/10.7554/eLife.30388.002

The following figure supplement is available for figure 1:

**Figure supplement 1.** NGL2 localizes to tips of horizontal cell axons, not rod bipolar cell dendrites.
DOI: https://doi.org/10.7554/eLife.30388.003

approximately ninefold (*Soto et al., 2013*; *Keeley et al., 2014*). Thus, multiple horizontal cells inner-vate most rods, which likely explains the remaining NGL2 staining at axon tips labeled by infection of single horizontal cells with AAV-sgNGL2-tdT. We conclude that our AAV-mediated CRISPR/Cas9 strategy removed NGL2 from horizontal cells with high efficiency (i.e., in 95% of infected cells).

## NGL2 regulates horizontal cell axon development cell autonomously

We first used this strategy, to analyze the effects of early postnatal NGL2 removal from individual horizontal cells on their development. In wild-type mice, P0 injection of AAV-sgNGL2-tdT affected neither the size of horizontal cell axons nor the density of their tips at P30 (*Figure 2A, B, F and G*). In Cas9 mice, axons of horizontal cells infected with AAV-sgNGL2-tdT were larger and had fewer tips than axons of horizontal cells infected with AAV-YFP (*Figure 2C–2G*). Both in wild-type mice and in Cas9 mice injected with AAV-sgNGL2-tdT, nearly all horizontal cell axon tips apposed ribbon release sites of rod photoreceptors (*Figure 2—figure supplement 1*). We, therefore, use axon tips throughout this study as an indicator of synapses between horizontal cells and rods. Changes in hori-zontal cell axon size and tip density in Cas9 mice injected with AAV-sgNGL2-tdT matched those observed in germline *Ngl2* knockout mice (*Ngl2$^{-/-}$* mice) (*Soto et al., 2013*). They were stable over time (*Figure 2—figure supplement 2*) and indistinguishable between both sgRNAs (*Figure 2C–2G*). Consistent with our findings in *Ngl2$^{-/-}$* mice (*Soto et al., 2013*), removal of NGL2 from individual hor-izontal cells did not affect the size of horizontal cell dendrites or the number of their contacts with cones (*Figure 2H–2N*). Thus, during development, NGL2 appears to selectively and cell-autono-mously regulate the growth of horizontal cell axons and the formation of synapses between horizon-tal cell axons and rods.

In *Ngl2$^{-/-}$* mice, horizontal cell axons frequently stray into the outer nuclear layer (*Soto et al., 2013*). Stray processes were less abundant in horizontal cells targeted by our AAV-mediated CRISPR/Cas9 strategy (*Figure 2—figure supplement 3*), indicating either that mistargeting involves non-cell-autonomous actions of NGL2, or that delays in the AAV-mediated removal of NGL2 reduced the severity of laminar targeting deficits compared to germline knockouts.

## NGL2 maintains horizontal cell axon tips and restrains axon growth in mature circuits

In mice, neuronal morphology and connectivity in the outer retina are mature 3–4 weeks after birth (*Huckfeldt et al., 2009*; *Poché et al., 2007*; *Blanks et al., 1974*). To test whether NGL2 contributes to the maintenance of horizontal cell axons and synapses, we injected AAV-sgNGL2-tdT in P30 (i.e., young adult) Cas9 mice. By P60, axon arbors of tdT-positive horizontal cells had expanded and lost many tips, compared to Cas9 mice injected with AAV-YFP (*Figure 3A–3E*). Axon expansion and tip loss were stable at P90 (*Figure 3—figure supplement 1*), indicating that the same phenotypes pro-duced by developmental NGL2 removal emerge rapidly and persist after removal of NGL2 in mature circuits.

Many neurons exhibit heightened plasticity during critical periods that can extend several weeks beyond their initial neurite growth and synaptogenesis (*Hong et al., 2014*; *Hensch, 2005*). To assess whether injection of AAV-sgNGL2-tdT at P30 fell within such a critical period or whether NGL2 con-tinues to be required throughout adulthood, we injected AAV-sgNGL2-tdT in mature adult (P150) Cas9 mice. Strikingly, by P180, horizontal cell axons had expanded and lost tips to the same degree observed for developmental and young adult manipulations (*Figure 4A–4E*). Injection of AAV-sgNGL2-tdT in P150 wild-type mice produced no phenotypes (*Figure 4—figure supplement 1*). Thus, NGL2 signaling appears to restrain horizontal cell axon growth and to maintain synapses between horizontal cells and rods throughout adulthood.

## AAV-mediated NGL2 expression in developing *Ngl2$^{-/-}$* horizontal cells normalizes axon growth and enhances axon tip density

Because removal of NGL2 from individual horizontal cells disrupted axon and synapse development, we wanted to test whether expression of NGL2 in individual horizontal cells in *Ngl2$^{-/-}$* mice could res-cue axon and synapse development. We generated AAVs expressing full-length NGL2 (AAV-NGL2) in horizontal cells and injected AAV-NGL2 or AAV-YFP intravitreally at P0. At P30, we counted axon tips of infected horizontal cells by immunostaining for NGL2 (AAV-NGL2) or YFP (AAV-YFP), and

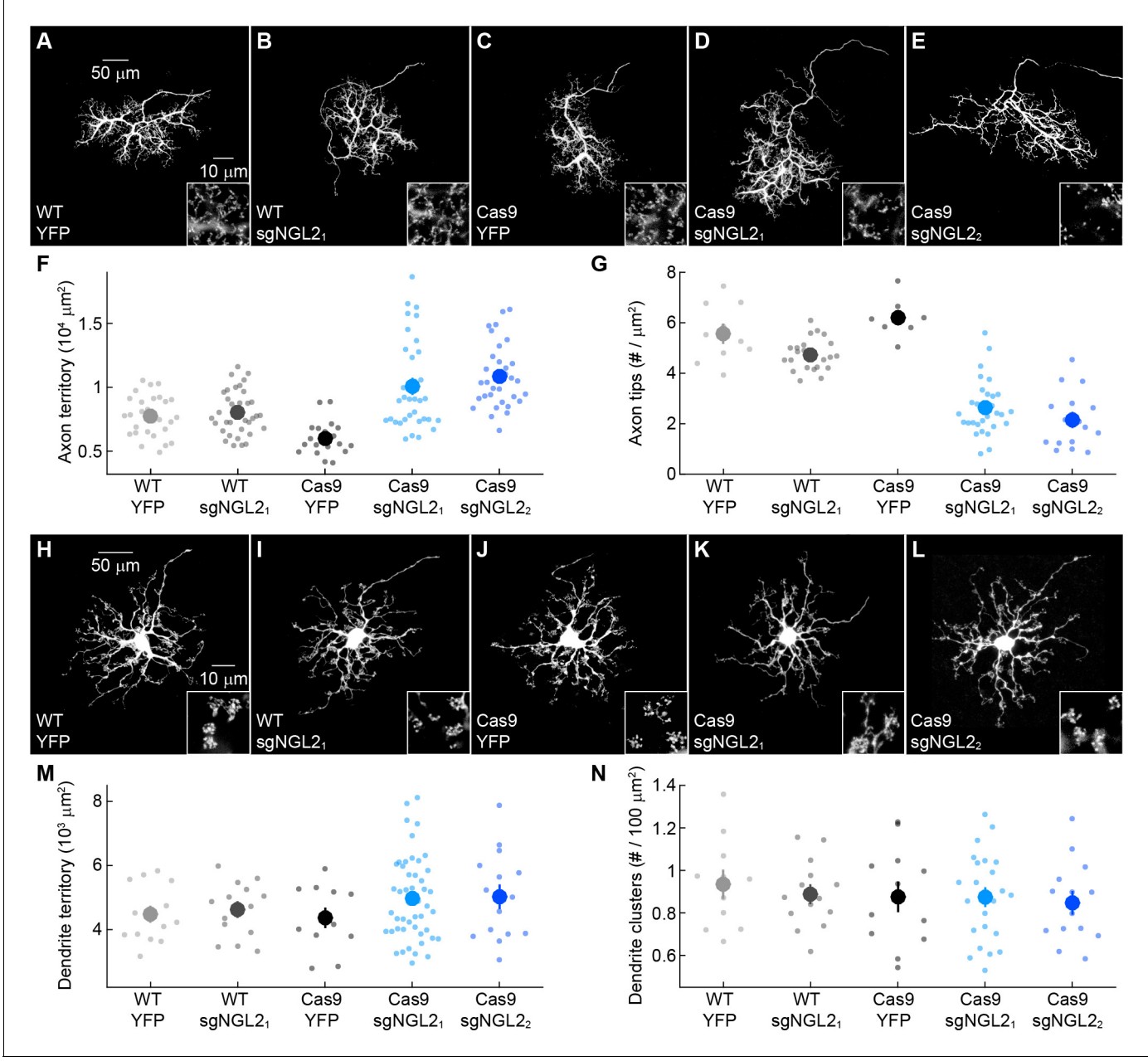

**Figure 2.** NGL2 regulates horizontal cell axon size and tip density cell autonomously. (**A–E**) Representative images of horizontal cell axons labeled by AAV-YFP (YFP) or AAV-sgNGL2-tdT (sgNGL2$_1$ and sgNGL2$_2$) in wild-type (WT) and Cas9 mice. Overview images are maximum intensity projections of the complete axons; insets show maximum intensity projections limited to axon tips at higher magnification. (**F and G**) Summary data of axon territories (**F**) and the density of axon tips in these territories (**G**). Dots show data from single cells, circles (errorbars) indicate means (±SEM) of the respective populations. AAV-sgNGL2-tdT infection in wild-type mice did not affect horizontal cell axon size (WT YFP n = 29, 8 mice, WT sgNGL2$_1$n = 37, 12 mice, p=1) or tip density (WT YFP n = 10, 3 mice, WT sgNGL2$_1$n = 23, 8 mice, p=0.25). Horizontal cell axons in Cas9 mice were similar in size (WT YFP n = 29, 8 mice, Cas9 YFP n = 20, 8 mice, p=0.11) and tip densities (WT YFP n = 10, 3 mice, Cas9 YFP n = 8, 4 mice, p=1) to horizontal cell axons in wild-type mice. Both sgRNAs tested drastically increased horizontal cell axon size in Cas9 mice (Cas9 sgNGL2$_1$n = 35, 6 mice, p<10$^{-7}$ for comparison to Cas9 YFP, Cas9 sgNGL2$_2$n = 32, 7 mice, p=10$^{-9}$ for comparison to Cas9 YFP) and reduced tip densities (Cas9 sgNGL2$_1$n = 29, 5 mice, p < 10$^{-12}$ for comparison to Cas9 YFP, Cas9 sgNGL2$_2$n = 16, 3 mice, p<0.001 for comparison to Cas9 YFP). (**H – L**) Analogous to (**A – E**) for horizontal cell dendrites and their contacts with cones. (**M and N**) Analogous to (**F and G**) for horizontal cell dendrites territories (WT YFP n = 15, 8 mice, WT sgNGL2$_1$n = 15 , 9 mice, Cas9 YFP n = 12, 8 mice, Cas9 sgNGL2$_1$n = 47, 6 mice, Cas9 sgNGL2$_2$n = 15, 7 mice) and terminal clusters within these territories (WT YFP n = 12, 6 mice, WT sgNGL2$_1$n = 15, 9 mice, Cas9 YFP n = 12, 8 mice, Cas9 sgNGL2$_1$n = 14, 3 mice, Cas9 sgNGL2$_2$n = 14, 6 mice). No significant differences between genotypes and AAVs were observed for horizontal cell dendrites (p>0.9, for all comparisons). P-values reported in this figure legend are from ANOVA tests with Bonferroni correction for multiple comparisons.

*Figure 2 continued on next page*

*Figure 2 continued*

DOI: https://doi.org/10.7554/eLife.30388.004

The following figure supplements are available for figure 2:

**Figure supplement 1.** Nearly all horizontal cell axon tips are synaptic.

DOI: https://doi.org/10.7554/eLife.30388.005

**Figure supplement 2.** Stable changes in horizontal cell axon size and axon tip density after NGL2 removal.

DOI: https://doi.org/10.7554/eLife.30388.006

**Figure supplement 3.** Effects of NGL2 removal on horizontal cell axon targeting.

DOI: https://doi.org/10.7554/eLife.30388.007

measured axon territories as the area of the smallest convex polygon to encompass all tips of an arbor. AAV-mediated expression of NGL2 exceeded wild-type protein levels at P30 (*Figure 5—figure supplement 1*). In *Ngl2*$^{-/-}$ mice, AAV-NGL2 restored the size of horizontal cell axons to wild-type

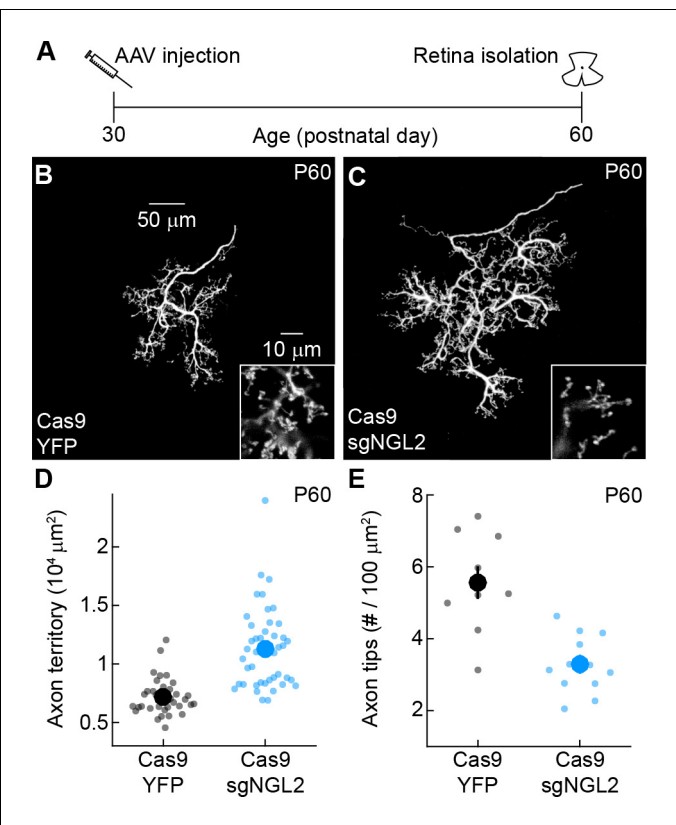

**Figure 3.** NGL2 restrains horizontal cell axon growth and maintains axon tips in young adult mice. (**A**) Schematic of the experimental timeline. AAVs were injected into the vitreous chamber of P30 mice and retinas collected on P60. (**B and C**) Representative images of horizontal cell axons labeled by AAV-YFP (YFP) or AAV-sgNGL2-tdT (sgNGL2) in Cas9 mice, imaged at P60. Overview images are maximum intensity projections of the complete axons; insets show maximum intensity projections limited to axon tips at higher magnification. (**D and E**) Summary data of axon territories (**D**) and the density of axon tips in these territories (**E**) at P60. Dots show data from single cells, circles (errorbars) indicate means (±SEM) of the respective populations. At P60, axons of horizontal cells infected with AAV-sgNGL2-tdT occupied larger territories (Cas9 YFP n = 35, 15 mice, Cas9 sgNGL2 n = 46, 6 mice, p<10$^{-9}$, Wilcoxon rank sum test), and had lower densities of axon tips (Cas9 YFP n = 10, 3 mice, Cas9 sgNGL2 n = 12, 4 mice, p<0.001, Wilcoxon rank sum test) compared to horizontal cells infected with AAV-YFP.

DOI: https://doi.org/10.7554/eLife.30388.008

The following figure supplement is available for figure 3:

**Figure supplement 1.** NGL2 restrains HC axon growth and maintains axon tips in adult mice.

DOI: https://doi.org/10.7554/eLife.30388.009

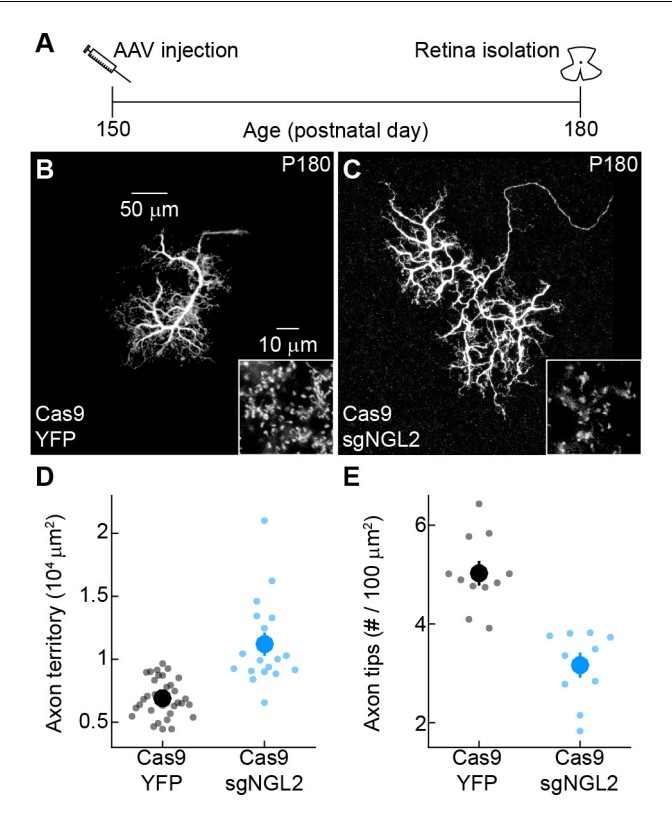

**Figure 4.** NGL2 restrains horizontal cell axon growth and maintains axon tips in mature adult mice. (**A**) Schematic of the experimental timeline. AAVs were injected into the vitreous chamber of P150 mice and retinas collected on P180. (**B and C**) Representative images of horizontal cell axons labeled by AAV-YFP (YFP) or AAV-sgNGL2-tdT (sgNGL2) in Cas9 mice. Overview images are maximum intensity projections of the complete axons; insets show maximum intensity projections limited to axon tips at higher magnification. (**D and E**) Summary data of axon territories (**D**) and the density of axon tips in these territories (**E**). Dots show data from single cells, circles (errorbars) indicate means (±SEM) of the respective populations. Axons of horizontal cells infected with AAV-sgNGL2-tdT occupied larger territories (Cas9 YFP n = 34, 9 mice, Cas9 sgNGL2 n = 18, 11 mice, $p<10^{-6}$, Wilcoxon rank sum test), and had lower densities of axon tips (Cas9 YFP n = 11, 3 mice, Cas9 sgNGL2 n = 10, 3 mice, $p<0.001$, Wilcoxon rank sum test) compared to horizontal cells infected with AAV-YFP.
DOI: https://doi.org/10.7554/eLife.30388.010

The following figure supplement is available for figure 4:

**Figure supplement 1.** Small guide RNAs targeting *Ngl2* do not affect horizontal cell axons in mature adult wild-type mice.
DOI: https://doi.org/10.7554/eLife.30388.011

levels and increased the density of axon tips beyond wild-type levels (*Figure 5A–5D, F and G*). In wild-type mice, AAV-NGL2 did not significantly change horizontal cell axon size but elevated axon tip density (*Figure 5D–5G*). The most parsimonious explanation for these findings is that a threshold amount of NGL2 is required to restrict horizontal cell axon growth and that NGL2 protein levels control synapse density bidirectionally. These results also further support the notion that NGL2 regulates horizontal cell development cell autonomously.

## AAV-mediated NGL2 expression in adult *Ngl2⁻ᐟ⁻* horizontal cells shrinks axons and restores axon tip density

We next explored whether restoring NGL2 to individual horizontal cells in adult *Ngl2⁻ᐟ⁻* circuits could reverse axonal and synaptic phenotypes. Indeed, injection of AAV-NGL2 at P30 reduced axon size and increased the density of axon tips of infected horizontal cells in *Ngl2⁻ᐟ⁻* mice to wild-type levels by P60 (*Figure 6A–D, F and G*). In wild-type mice, AAV-NGL2 injection at P30 did not significantly

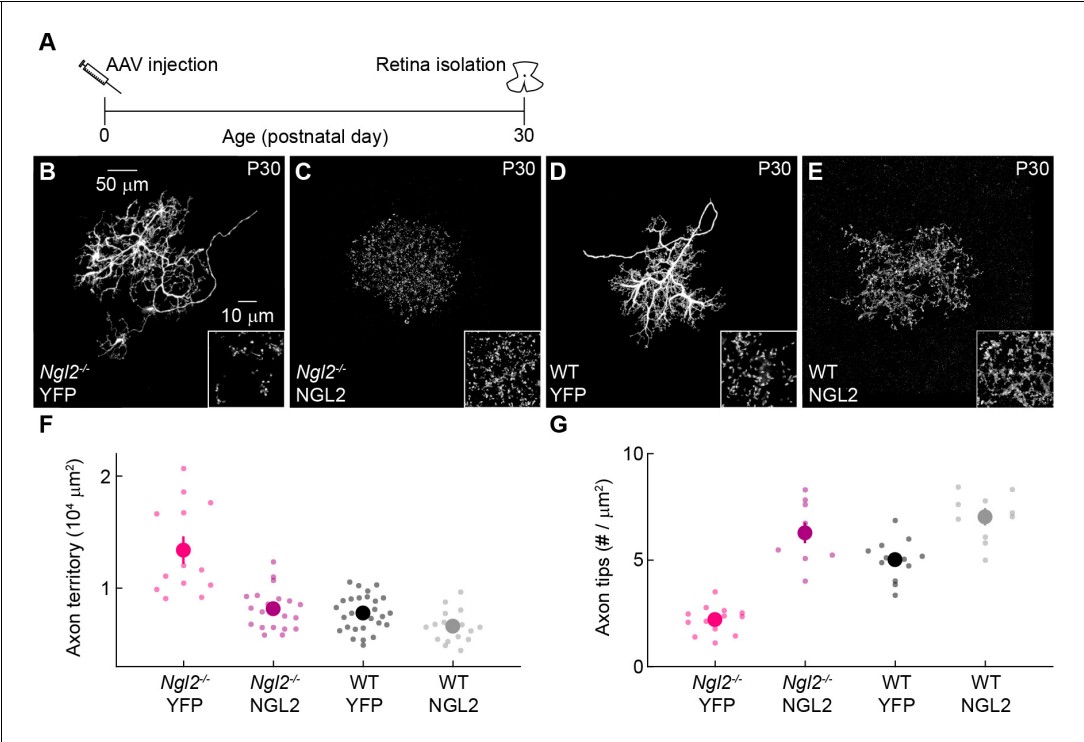

**Figure 5.** AAV-mediated NGL2 expression normalizes axon growth and enhances axon tip formation of horizontal cells in developing *Ngl2⁻/⁻* mice. (**A**) Schematic of the experimental timeline. AAVs were injected into the vitreous chamber of P0 mice and retinas collected at P30. (**B–E**) Representative images of axons of horizontal cells infected with AAV-YFP (YFP, (**B and D**) or AAV-NGL2 (NGL2, C, and E) in *Ngl2⁻/⁻* mice (**B and C**) or wild-type littermates (**D and E**). Overview images are maximum intensity projections of the complete axons; insets show maximum intensity projections limited to axon tips at higher magnification. (**F and G**) Summary data of axon territories (**F**) and the density of axon tips in these territories (**G**). Dots show data from single cells, circles (errorbars) indicate means (±SEM) of the respective populations. AAV-mediated expression of NGL2 reduced horizontal cell axon territories (*Ngl2⁻/⁻* YFP n = 12, 4 mice, *Ngl2⁻/⁻* NGL2 n = 21, 4 mice, p<10⁻⁷), restoring them to wild-type levels (WT YFP n = 31, 5 mice, p=1 for comparison to *Ngl2⁻/⁻* NGL2). In wild-type mice, axon territories of AAV-NGL2-infected horizontal cells were not significantly different from AAV-YFP-infected horizontal cells (WT NGL2, n = 15, 3 mice, p=0.48 for comparison to WT YFP). AAV-mediated NGL2 expression increased axon tip density in *Ngl2⁻/⁻* mice (*Ngl2⁻/⁻* YFP n = 10, 3 mice, *Ngl2⁻/⁻* NGL2 n = 12, 3 mice, p<10⁻¹⁰) beyond wild-type levels (WT YFP n = 12, 3 mice, p<10⁻⁶ for comparison to *Ngl2⁻/⁻* NGL2). AAV-mediated expression of NGL2 in wild-type mice similarly increased the axon tip density of horizontal cell axons (WT NGL2, n = 10, 3 mice, p<0.001 for comparison to WT YFP). P-values reported in this figure legend are from ANOVA tests with Bonferroni correction for multiple comparisons.

DOI: https://doi.org/10.7554/eLife.30388.012

The following figure supplement is available for figure 5:

**Figure supplement 1.** AAV-mediated NGL2 expression in *Ngl2⁻/⁻* and wild-type P30 retinas.
DOI: https://doi.org/10.7554/eLife.30388.013

change the axon size of infected horizontal cells but elevated the density of axon tips (*Figure 6D–6G*). AAV-mediated expression of NGL2 after P30 injections exceeded wild-type protein levels by P60 (*Figure 6—figure supplement 1*). Thus, re- and overexpression of NGL2 in mature horizontal cells shrink axons back to their normal size in *Ngl2⁻/⁻* mice, and can restore and even enhance their connectivity in *Ngl2⁻/⁻* and wild-type mice.

## Discussion

Synaptic CAMs like NGL2 typically localize to nascent connections during development and remain at synapses for the lifetime of the animal (*de Wit and Ghosh, 2014*; *Siddiqui and Craig, 2011*; *Sytnyk et al., 2017*). Because most in vivo studies have used knockout mice that remove targeted genes in the germline, it is unclear whether synaptic CAMs contribute to the maintenance of synapses as well as their development. Here, we devised an AAV-mediated CRISPR/Cas9 strategy to

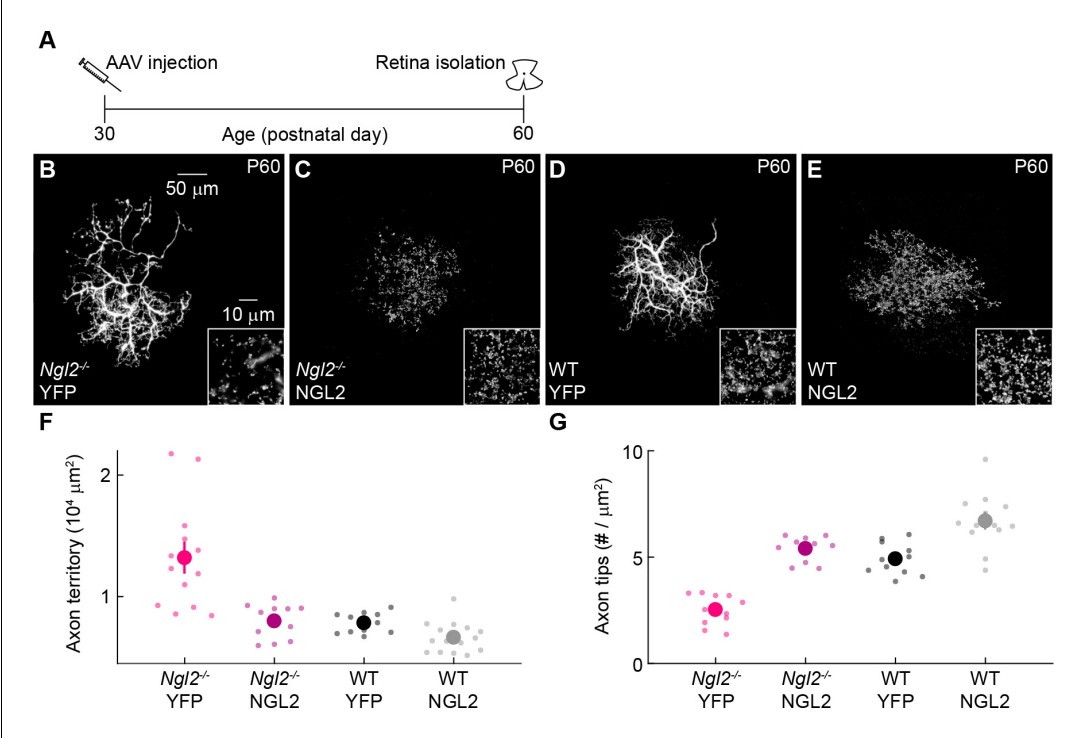

**Figure 6.** AAV-mediated NGL2 expression shrinks axon arbors and restores axon tips of horizontal cells in adult *Ngl2⁻/⁻* mice. (A) Schematic of the experimental timeline. AAVs were injected into the vitreous chamber of P30 mice and retinas collected at P60. (B–E) Representative images of axons of horizontal cells infected with AAV-YFP (YFP, (B and D) or AAV-NGL2 (NGL2, (C and E) in *Ngl2⁻/⁻* mice (B and C) or wild-type littermates (D and E). Overview images are maximum intensity projections of the complete axons; insets show maximum intensity projections limited to axon tips at higher magnification. (F and G) Summary data of axon territories (F) and the density of axon tips in these territories (G). Dots show data from single cells, circles (errorbars) indicate means (±SEM) of the respective populations. AAV-mediated expression of NGL2 reduced horizontal cell axon territories (*Ngl2⁻/⁻* YFP n = 12, four mice, *Ngl2⁻/⁻* NGL2 n = 12, two mice, p<10⁻⁴), restoring them to wild-type levels (WT YFP n = 11, four mice, p=1 for comparison to *Ngl2⁻/⁻* NGL2). In wild-type mice, axon territories of AAV-NGL2-infected horizontal cells were not significantly different from AAV-YFP-infected horizontal cells (WT NGL2, n = 15, four mice, p=1 for comparison to WT YFP). AAV-mediated NGL2 expression increased axon tip density in *Ngl2⁻/⁻* mice (*Ngl2⁻/⁻* YFP n = 11, 4mice, *Ngl2⁻/⁻* NGL2 n = 10, two mice, p<10⁻⁷) to wild-type levels (WT YFP n = 11, four mice, p=1 for comparison to *Ngl2⁻/⁻* NGL2). AAV-mediated expression of NGL2 in wild-type mice similarly increased the axon tip density of horizontal cells (WT NGL2, n = 13, four mice, p<0.001 for comparison to WT YFP). P-values reported in this figure legend are from ANOVA tests with Bonferroni correction for multiple comparisons.

DOI: https://doi.org/10.7554/eLife.30388.014

The following figure supplement is available for figure 6:

**Figure supplement 1.** AAV-mediated NGL2 expression in *Ngl2⁻/⁻* and wild-type P60 retinas.
DOI: https://doi.org/10.7554/eLife.30388.015

remove NGL2 from horizontal cells with temporal control (*Figure 1*). Developmental removal of NGL2 with this strategy reduced the density of synapses formed by horizontal cell axons to the same extent as *Ngl2⁻/⁻* mice (*Figure 2*) (*Soto et al., 2013*). When we removed NGL2 in one- and five-months-old mice, horizontal cell axons rapidly lost connections with rod photoreceptors (*Figures 3* and *4*). In the medial prefrontal cortex of adult mice, removal of neuroligin-2 reduces the number of inhibitory synapses (*Liang et al., 2015*). In the cerebellum, parallel fiber inputs are lost after deletion of *GluRδ2* in mature Purkinje cells (*Takeuchi et al., 2005*). Together, these findings indicate that synaptic CAMs stabilize synapses and that the cues that maintain circuits overlap with those that govern their formation.

Removal of NGL2 from individual horizontal cells (*Figure 2*) had the same effects on connectivity as removal from all cells in *Ngl2⁻/⁻* mice (*Soto et al., 2013*), and AAV-mediated expression of NGL2 in individual horizontal cells of *Ngl2⁻/⁻* mice was sufficient to rescue their axon and synapse development (*Figure 5*). Thus, NGL2 regulates horizontal cell connectivity cell autonomously. NGL2 appears to act similarly in the hippocampus, where knockdown in individual pyramidal cells lowers spine

density (*DeNardo et al., 2012*). The observation that AAV-mediated overexpression of NGL2 in horizontal cells in wild-type and *Ngl2*⁻/⁻ mice elevates axon tip density above wild-type levels (*Figure 5*) suggests that NGL2 levels control connectivity bidirectionally and limit synapse formation in wild-type retinas. Similarly, spine density in the hippocampus is reduced in *Cadm1*⁻/⁻ mice and increased in transgenic mice overexpressing SynCAM1, the protein encoded by *Cadm1* (*Robbins et al., 2010*). NGL2 interacts trans-synaptically with netrin-G2 (*DeNardo et al., 2012*; *Kim et al., 2006*; *Nishimura-Akiyoshi et al., 2007*; *Zhang et al., 2008*). Whether this interaction localizes NGL2 to horizontal cell axon tips, as it localizes NGL2 to proximal dendrites of CA1 pyramidal neurons (*Nishimura-Akiyoshi et al., 2007*), and how netrin-G2 – NGL2 complexes transmit signals between rods and horizontal cells remains to be explored. We know very little about the signaling mechanisms of synaptic CAMs, whether they act cell-autonomously or not, and whether they control neuronal processes (e.g., synapse formation) bidirectionally. Advances in viral targeting and genome editing (*Buchholz et al., 2015*; *Platt et al., 2014*; *Senís et al., 2014*; *Trapani et al., 2014*), which we exploited in our study, should accelerate progress in this area.

We found that NGL2 restricts horizontal cell axon growth throughout life (*Figures 2–4*). Similarly, the auxiliary Ca²⁺ channel subunit α2δ2, which mediates trans-synaptic interactions (*Fell et al., 2016*), restricts axon growth in developing and mature mouse dorsal root ganglion neurons (*Tedeschi et al., 2016*) and Ig-domain-containing CAMs contribute to axon maintenance in *C. elegans* (*Aurelio et al., 2002*; *Cherra and Jin, 2016*). Together, these results indicate that synaptic CAMs can control the size of neurite arbors during development and at maturity.

We found that manipulations of NGL2 co-regulate horizontal cell axon growth and synapse development in seemingly homeostatic fashion: deletion of *Ngl2* reduced synapse density and increased axons size (*Figures 2–4*), whereas re-expression of NGL2 in *Ngl2*⁻/⁻ mice increased synapse density and reduced axon size (*Figures 5* and *6*). Whether neurite growth and synapse formation in horizontal cells are coupled or independently controlled remains to be tested. A homeostatic relationship between these processes was reported for aCC neurons in *Drosophila* embryos (*Tripodi et al., 2008*). Bidirectional manipulations in the number of input synapses on aCC dendrites elicit opposite changes in arbor size (*Tripodi et al., 2008*). By contrast, neurite growth appears positively coupled to synapse formation of axons and dendrites in the optic tectum of *Xenopus* and Zebrafish, respectively (*Niell et al., 2004*; *Ruthazer et al., 2006*). In the mouse retina, the size and morphology of bipolar cell axons and ganglion cell dendrites are established and maintained independent of connections between them (*Johnson and Kerschensteiner, 2014*; *Kerschensteiner et al., 2009*; *Morgan et al., 2011*). What accounts for the differences in the relationship of synapses and neurite arbors in different neurons, circuits, and species, remains to be identified.

AAV-mediated expression of NGL2 in individual horizontal cells in adult *Ngl2*⁻/⁻ and wild-type mice restored and enhanced their connectivity (*Figure 6*). In neurodegenerative diseases, synapse loss can long precede cell death (*Buckingham et al., 2008*; *Gamm et al., 2015*; *Hong et al., 2016*; *Scheff et al., 2006*). Molecular mechanisms that maintain synapses could, therefore, be utilized to rescue circuit function in neurodegenerative diseases. We speculate that AAV-mediated expression of NGL2 might help preserve outer retinal connectivity in retinal degeneration, the most common heritable cause of visual impairment (*Sohocki et al., 2001*; *Jones et al., 2012*). Transplantation of stem-cell-derived photoreceptors is being explored as a treatment for late stages of retinal degeneration (*Gamm et al., 2015*; *Wahlin et al., 2017*; *Chirco et al., 2017*). Our results suggest that AAV-mediated expression of NGL2 in second-order neurons (i.e., bipolar and horizontal cells) could promote the integration of stem-cell-derived photoreceptors into host circuits and help restore vision.

# Materials and methods

### Key resources table

| Reagent type (species) or resource | Designation | Source or reference | Identifiers | Additional information |
|---|---|---|---|---|
| Mouse strain | Wild-type (C57Bl6/J) | Jackson Laboratory | RRID:IMSR_JAX000664 | |
| Mouse strain | Cas9 | Jackson Laboratory | RRID:IMSR_JAX:024858 | |

*Continued on next page*

*Continued*

| Reagent type (species) or resource | Designation | Source or reference | Identifiers | Additional information |
|---|---|---|---|---|
| Mouse strain | *Ngl2⁻/⁻* | (*Soto et al., 2013*) | | |
| Plasmid | *pX330-U6-Chimeric_BB-CBh-hSpCas9* | Addgene | Plasmid # 42230 | |
| Plasmid | *pmaxGFP* | Lonza | Catalog # VDC-1040 | |
| Virus | AAV-YFP | This paper | | |
| Virus | AAV-Grm6-YFP | (*Johnson et al., 2017*) | | |
| Virus | AAV-sgNGL2$_1$-tdT | This paper | | |
| Virus | AAV-sgNGL2$_2$-tdT | This paper | | |
| Virus | AAV-NGL2 | This paper | | |
| Antibody | anti-NGL2 (mouse) | NeuroMab | RRID:AB_2137614 | 1:100 dilution |
| Antibody | anti-Bassoon (mouse) | Enzo | RRID:AB_2313990 | 1:500 dilution |
| Antibody | anti-DsRed (rabbit) | Clontech Laboratories | RRID:AB_10013483 | 1:1000 dilution |
| Antibody | anti-GFP (chicken) | ThermoFisher | RRID:AB_2534023 | 1:500 dilution |
| Antibody | anti-GFP (rabbit) | ThermoFisher | RRID:AB_221569 | 1:1000 dilution |
| Antibody | anti-chicken IgY Alexa 488 | ThermoFisher | RRID:AB_2534096 | 1:1000 dilution |
| Antibody | anti-rabbit IgG Alexa 488 | ThermoFisher | RRID:AB_2536097 | 1:1000 dilution |
| Antibody | anti-rabbit IgG Alexa 568 | ThermoFisher | RRID:AB_143011 | 1:1000 dilution |
| Antibody | anti-mouse IgG Alexa 568 | ThermoFisher | RRID:AB_2534072 | 1:1000 dilution |
| Antibody | anti-mouse IgG Alexa 633 | ThermoFisher | RRID:AB_2535718 | 1:1000 dilution |

## Mice

In this study, we used mice in which the RNA-guided endonuclease Cas9 is ubiquitously expressed from a CAG promoter in the Rosa26 locus (Cas9 mice, Jackson Labs, RRID:IMSR_JAX:024858) (*Platt et al., 2014*). Cas9 mice were on a C57Bl6/N background. To confirm that sgRNAs had no effect in the absence of Cas9, we used wild-type mice on a C57Bl6/J background (RRID:IMSR_JAX000664). In addition, we used *Ngl2⁻/⁻* mice on a C57Bl6/J background. Mice of both sexes were used in our experiments. All procedures were approved by the Animal Studies Committee of Washington University School (protocol # 20170033) of Medicine and performed in compliance with the National Institutes of Health *Guide for the Care and Use of Laboratory Animals*.

## DNA constructs and sgRNA testing

Single guide RNAs targeting the 5' region of the mouse *Ngl2* gene were designed using an open access web tool provided by the Zhang lab at MIT (http://crispr.mit.edu:8079/) to search the first 250 bp of the *Ngl2* gene coding region. A pair of complementary oligonucleotides encoding the 20-nt guide sequences sgNGL2$_1$ (5'-gaatccacacttgcgccgtg-3') and sgNGL2$_2$ (5'-aaggtggtgtgcacccgccg-3') were synthetized. Oligonucleotides used in the generation of sgRNA templates were: Guide1-F 5'-ACCgaatccacacttgcgccgtg-3', Guide1-R 5'-AACcacggcgcaagtgtggattc-3', Guide2-F 5'-ACCaaggtggtgtgcacccgccg-3', and Guide2-R 5'-AACcggcgggtgcacaccacctt-3'.

Plasmid *pX330-U6-Chimeric_BB-CBh-hSpCas9* containing the U6 promoter, the sgRNA targeting sequence cloning site followed by the sgRNA scaffold, and the human codon-optimized Cas9 was obtained from Addgene (# 42230) and was used to clone the corresponding sgRNAs for testing in vitro. The *Ngl2⁻/⁻* cell line was generated from the mouse Neuro-2a line at the Genome Engineering and iPSC Center (GEiC), Washington University in St. Louis. Approximately $4 * 10^5$ Neuro-2a cells were suspended in P3 primary buffer and electroporated using a 4D-Nucleofector (Lonza) with 0.5 μg of *pmaxGFP* (control, Lonza) or *pX330-U6-Chimeric_BB-CBh*-hSpCas9 containing sgNGL2$_1$ or sgNGL2$_2$, in 20 μL wells. Sixty hours following nucleofection, cells were harvested, and genomic DNA was isolated and screened with PCRs using tagged primer sets (5'-CACTCTTTCCCTACAC-GACGCTCTTCCGATCTgctcctagctcacttaagccggggt-3', 5'-GTGACTGGAGTTCAGACGTGTGCTC-TTCCGATCTaggtgcctgaaggtgtcggcctgaa-3') specific to the targeted region. The targeted *Ngl2*

locus region was sequence-confirmed using next-generation sequencing. Approximately 1000 independent sequences were analyzed per plasmid used, and the percentage of non-homologous recombinant events detected was normalized to the total number of reads. Recombination rates were 60% and 68% for sgNGL2$_1$ or sgNGL2$_2$, respectively.

For constructing sgRNA-expressing AAV viral vectors, the original *pX330* plasmid was altered by adding a SapI site before the sgRNA scaffold. Then, the U6-sgRNA region of *pX330* was isolated with enzymes Acc65i-Afl III blunted with Klenow and cloned blunt upstream of the CAG promoter into an existing *pAAV-CAG-tdT* plasmid available in the lab. The new plasmid, *pAAV-U6-sgRNA-CAG-tdT*, was then digested with SapI to introduce the designed corresponding sgRNA oligos targeting *Ngl2*. For constructing the AAV viral vector expressing NGL2, full-length *Ngl2* cDNA cloned in *pEF-BOS* (*DeNardo et al., 2012*) was amplified using EcoRI and NotI linkers and cloned using the same sites into the *pAAV-CAG-YFP* vector in place of YFP. The PCR product was verified by sequencing.

## Adeno-associated viruses

Viral particles were packaged and purified as previously described (*Grimm et al., 2003*; *Klugmann et al., 2005*). Briefly, AAV1/2 chimeric virions, which readily infect horizontal cells (*Soto et al., 2013*), were produced by co-transfecting HEK293 cells with *pAAV-U6-sgNGL2$_1$-CAG-tdT* (AAV-sgNGL2$_1$-tdT), *pAAV-U6-sgNGL2$_2$-CAG-tdT* (AAV-sgNGL2$_2$-tdT), *pAAV-CAG-YFP* (AAV-YFP), *pAAV-CAG-NGL2* (AAV-NGL2) or *pAAV-Grm6-YFP* (AAV-Grm6-YFP) (*Johnson et al., 2017*) and helper plasmids encoding Rep2 and the Cap for serotype one and Rep2 and the Cap for serotype 2. Forty-eight hours after transfection, cells and supernatant were harvested and viral particles purified using heparin affinity columns (Sigma, Saint Louis, MO). Viruses (250 nL) were injected with a Nanoject II (Drummond) into the vitreous chamber of newborn mice anesthetized on ice

## Tissue preparation

Mice were deeply anesthetized with $CO_2$, killed by cervical dislocation, and enucleated. Retinas were isolated in oxygenated mouse artificial cerebrospinal fluid (mACSF$_{HEPES}$) containing (in mM): 119 NaCl, 2.5 KCl, 1 NaH$_2$PO$_4$, 2.5 CaCl$_2$, 1.3 MgCl$_2$, 20 HEPES, and 11 glucose (pH adjusted to 7.37 using NaOH) and mounted flat on black membrane discs (HABGO1300, Millipore, Burlington, MA), or left in the eyecup for 30 min fixation with 4% paraformaldehyde in mACSF$_{HEPES}$.

## Immunohistochemistry

After blocking for 2 hr with 5% Normal Donkey Serum in PBS, vibratome slices (thickness: 60 μm) embedded in 4% agarose (Sigma) were incubated overnight at 4°C with primary antibodies. Slices were then washed in PBS (3 × 20 min) and incubated with secondary antibodies for 2 hr at room temperature (RT). Flat-mount preparations were frozen and thawed three times after cryoprotection (1 hr 10% sucrose in PBS at RT, 1 hr 20% sucrose in PBS at RT, and overnight 30% sucrose in PBS at 4°C), blocked with 5% Normal Donkey Serum in PBS for 2 hr, and then incubated with primary antibodies for 5 d at 4°C and washed in PBS (3 × 1 hr) at RT. The following primary antibodies were used in this study: mouse anti-NGL2 (1:100, NeuroMab, Davis, CA, RRID:AB_2137614), mouse anti-Bassoon (1:500, Enzo, Farmingdale, NY, RRID:AB_2313990), rabbit anti-DsRed (1:1000, Clontech Laboratories, RRID:AB_10013483), chicken anti-GFP (1:500, ThermoFisher, Waltham, MA, RRID:AB_2534023), and rabbit anti-GFP (1:1000, ThermoFisher, RRID:AB_221569). Subsequently, flat mounts were incubated 1 d at 4°C with Alexa 488 (1:1000, ThermoFisher, anti-chicken IgY, RRID:AB_2534096, anti-rabbit IgG RRID:AB_2536097), Alexa 568 (1:1000, ThermoFisher, anti-rabbit IgG, RRID:AB_143011, anti-mouse IgG, RRID:AB_2534072), and Alexa 633 (1:1000, ThermoFisher, anti-mouse IgG, RRID:AB_2535718) secondary antibodies for and washed in PBS (3 × 1 hr) at RT.

## Imaging

Image stacks were acquired on an Fv1000 laser scanning confocal microscope (Olympus) with 20 × 0.85 NA and 60 × 1.35 NA oil immersion objectives at a voxel sizes of 0.206 μm – 0.3 μm (x/y – z axis) and 0.082 μm – 0.3 μm (x/y – z axis), respectively, or on a Zeiss 810 laser scanning confocal

microscope with an Airyscan detector through a 63 × 1.4 NA objective at a voxel size of 0.043 µm – 0.1 µm (x/y – z axis).

## Image analysis

Horizontal cell axon and dendrite territories were defined as the areas of the smallest convex polygons to encompass the respective arbors (*Figures 1–4*) or all synaptic tips (*Figures 5* and *6*) in z-projections of confocal image stacks acquired in retinal flat mounts, and were measured using Fiji (*Schindelin et al., 2012*). Horizontal cell dendrite clusters at cone terminals and horizontal cell axon tips, which penetrate rod spherules (*Peichl and González-Soriano, 1994*; *Wässle, 2004*), were identified by eye in confocal image stacks and their positions (x, y, and z) noted to count synapses and analyze their stratification (*Soto et al., 2013*). To evaluate the efficiency of NGL2 removal in horizontal cells infected with *AAV-sgNGL2-tdT*, we acquired confocal images of their axons in retinas stained for NGL2. The intensity of NGL2 labeling in rod spherules penetrated by infected horizontal cells was then compared to the NGL2 signal in neighboring rod spherules. For visual clarity, in some of the representative images, horizontal cell axons and dendrites were digitally isolated using Amira (ThermoFisher) to remove Mueller glia labeled within the field of view.

## Statistics

A power analysis using G*Power (*Faul et al., 2009*) on our initial data sets of Cas9 retinas injected with AAV-YFP and AAV-sgNGL2-tdT suggested that given the observed effect sizes for horizontal cell axon territories and synapses, total sample sizes exceed 16 for all comparisons between groups to achieve power of 0.95 at p<0.05. In our final data set, total sample sizes for all comparisons ranged from 18 to 71. To assess the statistical significance of differences between two groups, we used Wilcoxon rank sum for continuous quantitative data and $X^2$ tests for frequency observations of categorical data. For comparisons of quantitative data between multiple groups, we used ANOVA tests with Bonferroni correction for multiple comparisons.

# Acknowledgements

We thank members of the Kerschensteiner lab for helpful comments and suggestions throughout this study. We thank Michael Casey of the Vision Core at Washington University School of Medicine for help with the design of sgRNAs and the construction of plasmids. We acknowledge the Genome Engineering and iPSC Center (GEiC) at Washington University School of Medicine for validating sgRNAs in vitro. We are grateful to Drs. A Ghosh and L DeNardo for providing us with the mouse *Ngl2* cDNA. This work was supported by the National Institutes of Health (EY027411 to FS and DK, EY026978 and EY023341 to DK, and the Vision Core Grant EY0268), and by the Research to Prevent Blindness Foundation via an unrestricted grant to the Department of Ophthalmology and Visual Sciences at Washington University School of Medicine.

# Additional information

### Funding

| Funder | Grant reference number | Author |
| --- | --- | --- |
| National Eye Institute | R01EY027411 | Florentina Soto<br>Daniel Kerschensteiner |
| National Eye Institute | EY0268 | Daniel Kerschensteiner<br>Florentina Soto |
| National Eye Institute | R01EY026978 | Daniel Kerschensteiner |
| National Eye Institute | R01EY023341 | Daniel Kerschensteiner |

The funders had no role in study design, data collection and interpretation, or the decision to submit the work for publication.

## Author contributions

Florentina Soto, Conceptualization, Formal analysis, Supervision, Funding acquisition, Investigation, Visualization, Writing—original draft, Writing—review and editing; Lei Zhao, Investigation, Writing—review and editing; Daniel Kerschensteiner, Conceptualization, Formal analysis, Supervision, Funding acquisition, Visualization, Writing—original draft, Writing—review and editing

## Author ORCIDs

Daniel Kerschensteiner http://orcid.org/0000-0002-6794-9056

## Ethics

Animal experimentation: All procedures were approved by the Animal Studies Committee of Washington University School (protocol # 20170033) of Medicine and performed in compliance with the National Institutes of Health Guide for the Care and Use of Laboratory Animals.

## Decision letter and Author response

Decision letter https://doi.org/10.7554/eLife.30388.018
Author response https://doi.org/10.7554/eLife.30388.019

# Additional files

**Supplementary files**
• Transparent reporting form
DOI: https://doi.org/10.7554/eLife.30388.016

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
