## [Decision Letter]

Thank you for submitting your article "NGL2 restrains axon growth and maintains synapses of mature retinal horizontal cells" for consideration by *eLife*. Your article has been reviewed by three peer reviewers, and the evaluation has been overseen by a Reviewing Editor and K VijayRaghavan as the Senior Editor. The reviewers have opted to remain anonymous.

The reviewers have discussed the reviews with one another and the Reviewing Editor has drafted this decision to help you prepare a revised submission.

Your study investigating the role of the synaptic cell adhesion molecule NGL2 in adult retinal horizontal cells (HCs) using a CRISPR/Cas9 genome engineering strategy to excise NGL2 in postnatal animals and asking if NGL2 is required 1) cell autonomously in horizontal cells for constraint of their axonal arbor development, and 2) for maintenance of horizontal cell – rod photoreceptor synapses in adult animals, has been reviewed rather positively. The reviewers and the Reviewing Editor thought that the evidence for synapse formation and maintenance and axon formation as processes that exclude each other were significant and exciting.

However, the reviewers' positive comments are offset by requests for analyses and controls that we feel would strengthen the claims of the paper. There are three categories of revisions; the three categories are viewed as generally essential.

1) A major criticism brought up by all three reviewers is the naming of axonal tips "synapses". Axon tips are used as a proxy for synapses throughout this study but bona fide synapses or synaptic markers other than NGL2 were not used to verify that the tips are synapses.

At a minimum the wording should be changed throughout the paper to reflect the fact that synapses were not analyzed, and co-staining with a presynaptic marker (synapsin, synaptophysin) should be performed. Ideally, the main conclusions of the paper should be bolstered by examining synaptic contacts after adult, conditional loss of NGL2, using immunohistochemistry and/or immuno-EM to analyze synapses (or lack thereof) in GFP+ axons. And, it would be important to demonstrate how adult loss of NGL2 affects axon tips contacting rods versus axon tips not contacting rods.

In line with considering the synapse issue: although the fact that loss of one adhesion molecule could cause a synapse to "dissolve" and axons to regrow is surprising, the importance of this finding is not well addressed in the text, nor are possible models described. The role of the binding partner for NGL2 – Netrin-G2 – could also be discussed. These last three points could be amended textually.

2) Off-target effects and rescue experiments: While the sgRNA approach to ablate NGL2 in the Cas9 transgenic mice is welcome, the reviewers suggested re-expressing NGL2 in the NGL2 ablated horizontal cells as a control for the specificity of the guides. The phenotype would be predicted to revert to the wild type condition. This would be very challenging and you would need to generate a second AAV that expresses NGL2 and achieve co-infection. Alternatively, one could use the CRISPR deletion method in NGL2 KO mice to see if no further effects were apparent.

3) More careful consideration of NGL2 expression to confirm loss of NGL2 protein: From the light-level assessment of "co-localization", both in this study and your 2013 study, it is not clear how the associations defined here are only postsynaptic in HCs, and the reviewers request some additional ICC experiments to support this point. Comments and possible amendments, in particular, from reviewer 3 include:a) That it is not clear that NGL2 expression was thoroughly validated in the 2013 study since sense controls were not presented along with the ISH experiments that placed NGL2 in horizontal cells, and antibody specificity was not validated on retinal sections or whole mount retinas from NGL2-/- animals.

b) That TdTomato expression should itself be a proxy of CRISPR knockout of NGL2 in the targeted cell, but additional analysis would help rule out the possibility that NGL2 has not been removed from neighboring bipolar cells (BPs), where it might lead to the phenotypes described here.

c) Use of a BP cell marker (perhaps PKCalpha and/or mGluR6) for imaging (high resolution confocal imaging should be tried) with NGL2 and TdTomato here would provide additional support for your conclusions.

d) Performing NGL2 staining of an AAV-YFP horizontal cell axon arbor control, and its quantification, to be able to compare it with the presented CRISPR-targeted NGL2 knockout cell in Figure 1.

4) Citations and relating your story to other published work: Tedeschi et al., Neuron, 2016, as mentioned by reviewer 2, describe that synapse formation and axon growth may be events that exclude one another, in a regeneration scenario, but nonetheless should be cited. There is also an interesting literature from the Hobert lab on *C. elegans* and axon/synapse maintenance, first published over 10 years ago (Aurelio et al., Science 2002), and recently by Y. Jin's lab (Neuron, 2016 PMID: 26796686, that might be relevant.

Should you be able to acknowledge that the tips of the axons were not verified as synapses (and/or do the minimum to verify synapses), and sufficiently amend #2 on rescue and #3 on protein validation in two months, and the study is deemed appropriate for publication, then the hard-core validation of synapse structure using immune-EM could be submitted and hopefully published as an "Advance" (see https://elifesciences.org/articles/research-advance). The new *eLife* Research Advance mechanism is a short article that allows the authors of an *eLife* paper to publish new results that build on their original research paper in an important way.

*Reviewer #1:*

The manuscript by Soto et al. investigates the role of the synaptic cell adhesion molecule NGL2 in adult retinal horizontal cells (HCs). The authors use a clever CRISPR method to delete NGL2 from sparse HCs at various time points after early development. The data presented here show that NGL2 cell autonomously constrains HC axon growth while positively regulating synapse-forming structures in mature neurons. Unfortunately, I have two major concerns with the present study.

1) The data are overall of good quality but the new knowledge gained from this study only incrementally advances results reported in their more extensive 2013 J. Neuroscience paper. Essentially, the new study shows that NGL2 has the same functions in young and old neurons. Advancing the field more substantially will require more experiments in a new direction, possibly investigation of NGL2 signaling or ligands/receptors during these processes.

2) The authors exclusively use axon tips as a proxy for synapses throughout this study but they never actually analyze bona fide synapses or synaptic markers other than NGL2. They often state "synapse tips" in the text and figure legends when "axon tips" should be stated. In their previous paper, they show that about 75% of axon tips contact rod photoreceptors. Here, it is not actually shown how HC-rod synapses may be affected by loss of NGL2 in adult neurons. How does adult loss of NGL2 affect axon tips contacting rods versus axon tips not contacting rods? One can imagine that existing HC-rod synaptic contacts may be more stable and harder to disperse than HC axon tips not in contact with rods. The main conclusions of the paper would be much stronger if true synapse contacts were examined after adult, conditional loss of NGL2, possibly using immuno-EM methods.

*Reviewer #2:*

The manuscript "NGL2 restrains axon growth and maintains synapses of mature retinal horizontal cells" by Soto et al. reports that deletion of the cell adhesion molecule NGL2 in individual retinal horizontal cells increases axon length and reduces their synaptic connections. Specifically, the authors used two sets of small guiding RNAs (sgRNAs) that were brought into HCs through adeno-associated virus (AAV) mediated delivery using transgenic mice that express Cas9 endonuclease ubiquitously. Injection of AAV containing sgRNAs for NGL2 into the vitreous chamber at postnatal day 1 (P1) led to longer axons and less synaptic structures at P30. The authors then investigated whether ablation of NGL2 at later stages (P30), when neuronal circuitry is well established, still leads to changes in axon harbors and found axon length increases and synaptic structures decrease in horizontal cells at P60. Similar results were also obtained in fully adult mice (P150).

The presented work from the Kerschensteiner lab is truly exciting. Their study demonstrates that a synaptic cell adhesion molecule has a dual function to enable functional circuitry: it restrains axon growth and it promotes synapse formation. It provides exciting genetic evidence that synapse formation maintenance and axon formation could be processes that exclude each other. The paper has certainly the potential to be an *eLife* paper. Before publication the authors should perform the following experiments and work on the manuscript on several aspects. I mention relatively few points and would expect that they will be done.

1) Potential off-target effects: The sgRNA approach to ablate NGL2 in the Cas9 transgenic mice is very nice. However, the authors should try re-expressing NGL2 in the NGL2 ablated horizontal cells. Is the phenotype then reverted to wild type condition?

2) Synapses: If the reviewer understands the study correctly, the authors did not use synaptic markers but focus on axon tips when they present their results on synapse reduction in NGL2-ablated horizontal cells. Can the authors perform a co-staining with a presynaptic marker (synapsin, synaptphysin)?

3) Statistics: Can the authors write a bit more detailed regarding the statistical tests they used? When they have multiple groups, do they use the Bonferoni corrections?

4) A recent paper (Tedeschi et al., Neuron, 2016) proposed that axon growth and synapse formation may be cellular events that exclude each other. The authors should discuss their findings in the context with this paper.

*Reviewer #3:*

Following up on their previous study describing functions of NGL-2 in horizontal cell (HC) neurite lamination, axon arbor elaboration, and synaptic assembly during early postnatal retinal development (Soto et al., 2013), the authors here examine the requirement for NGL2 in neonatal, young adult and mature adult mice for the connectivity of mature retinal circuits. They employed a CRISPR/Cas9 genome engineering strategy to excise NGL2 in postnatal animals and ask if it is required 1) cell autonomously in horizontal cells for constraint of their neurite lamination and axonal arbor development, and 2) for the maintenance of horizontal cell – rod photoreceptor synapses in adult animals. The results are quite interesting and strongly suggest that NGL2 is important not just during neonatal developmental time points for elaboration of HC axon/rod contacts, but also for their maintenance long into adulthood.

The chosen CRISPR/Cas9 strategy is appropriate and state-of-the-art, and aspects of the observed phenotypes are striking. However, this study could benefit from a more careful consideration of NGL2 expression and knockdown in targeted horizontal cells in order to support the authors' hypothesis that cell autonomous NGL2 restrains horizontal cell axon growth. If the authors are able to address the following points, study would be greatly strengthened and a good candidate for publication in *eLife*.

1) The present paper depends critically upon light-level assessment of NGL-2 expression and axon tips from CRISPR-manipulated and non-CRISPR-manipulated GFP-expressing HCs, and assumptions about how this association defines synaptogenesis between HC axons and rod photoreceptors. However, the previous study by these investigators (Soto et al., 2013) upon which this present study is based leaves open a few questions that would be very helpful for the authors to address here in order to strengthen this study. First, is not clear that NGL2 expression was thoroughly validated in the 2013 study since sense controls were not presented along with the ISH experiments that placed NGL2 in horizontal cells, and antibody specificity was not validated on retinal sections or whole mount retinas from NGL2-/- animals. These two simple experiments should be done to support this present communication.

Further, if the NGL2 protein signal clearly overlapped with TdTomato+ horizontal cell axon tips, it would lend strength to the argument that NGL2 is indeed expressed presynaptically at horizontal cell axon/rod photoreceptor synapses. However, the signal in panel D from Figure 2 of Soto et al., 2013 does not appear to overlap with the labeled horizontal cell. Rather, the signal appears juxtaposed with the TdTomato signal, calling into question whether or not it actually arises from co-labeling in HC axons. This NGL2 labeling pattern adjacent the axonal arbor of an otherwise intact, wild type, HC observed in Soto et al., 2013, Figure 2, is reminiscent of the signal presented in the current study for a proposed NGL2 CRISPR knockout horizontal cell (Figure 1), which weakens the argument that NGL2 protein is absent in the targeted horizontal cell because of the proposed CRISPR-mediated NGL2 excision event. Although TdTomato expression should itself be a proxy of CRISPR knockout of NGL2 in the targeted cell, additional would help rule out the possibility that NGL2 has not been removed from neighboring bipolar cells (BPs), where it might lead to the phenotypes described here. Though weak, in the 2013 paper there is some NGL2 in situ in the INL at P10 and increasingly so at P20 (adult in situs were not presented in Soto et al., 2013). Therefore, use of a BP cell marker (perhaps PKCalpha and/or mGluR6) for imaging (high resolution confocal imaging should be tried) with NGL2 and TdTomato here would provide support for the authors' conclusions. In addition, some explanation for how the non-overlapping but adjacent HC axon/axon tip labeling of TdTomato /NLG2 aligns with the authors' contention that these are both labeling the same presynaptic sites should be provided.

2) Related to the point raised above, it is problematic to refer categorically to "fewer axon tips," in the absence of a thorough characterization of synaptic structure and function (as was performed in Soto et al., 2013), as a proxy for "fewer synapses." References in the text to "fewer synapses" should be toned down to indicate fewer axon tips, or at least some qualification should be mentioned, unless of course additional synaptic markers are included in these analyses.

3) It is critical to see NGL2 staining of an AAV-YFP horizontal cell axon arbor control, and its quantification, so as to be able to compare it with the presented CRISPR-targeted NGL2 knockout cell in Figure 1. In addition, one assumes the non-Cas9 WT young and mature adult mice show none of the phenotypes indicated here for the Cas9 adult mice.

4) Figure 2, describing HC axon contacts with cones, appear not to be discussed in the body of the Results section. Please include referral to these data in the main text.

5) The conclusion that "mistargeting in the germline knockout involves non-cell-autonomous actions of NGL2" requires qualification. It seems quite plausible that HC axon growth leading to stray processes could result from incomplete knock down of NGL2, and indeed these data presented here showing reduced, but not completely absent, NGL-2 association with axon tips from sgRNA-expressing HCs support this idea.

[Editors' note: further revisions were requested prior to acceptance, as described below.]

Thank you for resubmitting your work entitled "Maintenance and restoration of connectivity in the retina by NGL2" for further consideration at *eLife*. Your revised article has been favorably evaluated by K VijayRaghavan (Senior Editor), a Reviewing Editor, and three reviewers.

The manuscript has been improved but there are some remaining issues that need to be addressed before acceptance, as outlined below:

All three reviewers found your revised manuscript substantially improved by the addition of new data showing that re-expression of NGL2 restores axon morphology in NGL2-null axons. This is an exciting new finding that simultaneously acts as a control for off-target crispr guide effects and suggests axon deficits can have potential to be reversed during disease, in the mature nervous system.

Nonetheless, one of the reviewers cited some remaining issues that need to be addressed before acceptance, as outlined below:

1) Considering, and naming of, axon tips as synapses: You have nicely demonstrated that horizontal cell axon tips almost always end in a synapse as indicated by the presence of the presynaptic marker bassoon; this validates their use of axon tips as a proxy for synapses. However, in the remainder of the paper, you still count axon tips and not synapses; thus, it would be more accurate to refer to the axon tips as such and not as "synapses" or "synapse tips". This consideration is similar to counting spines. In the literature, one would not count spines for an experiment and then state that synapse density is changed. You would still state that spine density is changed, which likely reflects altered synapse number.

Also, the Materials and methods do not list the bassoon antibody, its source, etc., nor describe the details of the staining experiment; please amend.

2) NGL2 expression in WT: In the NGL2 expression experiments (Figure 5 and Figure 6), did the virally expressed NGL2 have an epitope tag? If not, how were the NGL2-expressing cells identified in WT mice? Staining with the NGL2 antibody in WT mice should reveal a sea of NGL2 puncta. How were the images in 5E and 6E obtained? If a subtraction erasing of signal using Amira software was used as alluded to in the Materials and methods, the original images should be shown in a supplement. It is still unclear how they identified the overexpressing neurons because the images in the supplements for Figure 5 and Figure 6 are insufficient to support the stated conclusion that "AAV-mediated expression of NGL2 exceeded WT protein levels". WT levels should be shown for comparison and if possible, some kind of quantification is warranted or the statement revised.

3) Statistics: The authors indicate that they conducted multiple comparisons after each ANOVA and p-values are often listed after particular testing conditions in the figure legends. However, identification of the groups the p-value describes is usually not present. For example, are the p values for the ANOVA or for some post-hoc multiple comparison? If a multiple comparison, which groups? It would be easier to read and quickly interpret if the p-values were indicated in the graphs themselves or in a separate chart as a supplement. This comment pertains to most of the figures. Finally, we recommend that you provide more information about the statistics as described in comment above.

4) Graph in Figure 1 has no x-axis label.

5) Reference to Figure 2 is not referred in the text, nor any explanation of the experiment provided.

---

## [Author Response]

[…] However, the reviewers' positive comments are offset by requests for analyses and controls that we feel would strengthen the claims of the paper. There are three categories of revisions; the three categories are viewed as generally essential.1) A major criticism brought up by all three reviewers is the naming of axonal tips "synapses". Axon tips are used as a proxy for synapses throughout this study but bona fide synapses or synaptic markers other than NGL2 were not used to verify that the tips are synapses.At a minimum the wording should be changed throughout the paper to reflect the fact that synapses were not analyzed, and co-staining with a presynaptic marker (synapsin, synaptophysin) should be performed. Ideally, the main conclusions of the paper should be bolstered by examining synaptic contacts after adult, conditional loss of NGL2, using immunohistochemistry and/or immuno-EM to analyze synapses (or lack thereof) in GFP+ axons. And, it would be important to demonstrate how adult loss of NGL2 affects axon tips contacting rods versus axon tips not contacting rods.In line with considering the synapse issue: although the fact that loss of one adhesion molecule could cause a synapse to "dissolve" and axons to regrow is surprising, the importance of this finding is not well addressed in the text, nor are possible models described. The role of the binding partner for NGL2 – Netrin-G2 – could also be discussed. These last three points could be amended textually.

Following the reviewers’ suggestion, we stained for the presynaptic ribbon anchoring protein Bassoon to test what fraction of horizontal cell axon tips are synaptic in control conditions and after removal of NGL2. Both in control conditions (96.7% ± 1.3%, n = 7) and after removal of NGL2 (97.7% ± 1.5%, n = 4) nearly all horizontal cell axon tips were apposed by presynaptic ribbons of rod photoreceptors. We conclude that axon tips can be used to count synapses between horizontal cells and rod photoreceptors, and that the reduced density of axon tips caused by removal of NGL2 reflects synapse loss. We show results from the Bassoon staining in a new figure supplement (Figure 2—figure supplement 1) and explain our use of tips as a proxy for synapses in the Results section of our revised manuscript.

Furthermore, in our revised manuscript, we have tried to clarify the importance of our results regarding the role of NGL2 in synapse formation and stability, and axon growth and maintenance; we discuss the potential role of Netrin-G2 in regulating horizontal cell axon growth and the maintenance of synapses between horizontal cells and rod photoreceptors.

2) Off-target effects and rescue experiments: While the sgRNA approach to ablate NGL2 in the Cas9 transgenic mice is welcome, the reviewers suggested re-expressing NGL2 in the NGL2 ablated horizontal cells as a control for the specificity of the guides. The phenotype would be predicted to revert to the wild type condition. This would be very challenging and you would need to generate a second AAV that expresses NGL2 and achieve co-infection. Alternatively, one could use the CRISPR deletion method in NGL2 KO mice to see if no further effects were apparent.

We generated new AAVs to express NGL2 in individual horizontal cells in NGL2^-/-^ and wild-type mice. AAVmediated NGL2 expression exceeded wild-type levels. Restoring NGL2 to individual horizontal cells in NGL2^-/-^ mice restored axon size of infected cells to wild-type levels and increased synapse density beyond wild-type levels. This was true irrespective of whether AAVs were injected at P0 (i.e. before phenotype develops in NGL2^-/-^ mice) or at P30 (i.e. after phenotype develops in NGL2^-/-^ mice). Overexpression of NGL2 in individual horizontal cells in wild-type mice increased synapse density in infected compared to uninfected cells. This was observed both for injections at P0 and P30. These results, which we present in Figure 5 and Figure 6 of our revised manuscript, not only support the specificity of our sgRNA approach, but also demonstrate that NGL2 can drive synapse formation and axon retraction in developing and mature horizontal cells. Combining sgRNA knockout and AAV-mediated rescue and overexpression, we thus find that NGL2 levels restrict axon size and control synapse density in developing and mature retinal circuits in a bidirectional manner.

3) More careful consideration of NGL2 expression to confirm loss of NGL2 protein: From the light-level assessment of "co-localization", both in this study and your 2013 study, it is not clear how the associations defined here are only postsynaptic in HCs, and the reviewers request some additional ICC experiments to support this point. Comments and possible amendments, in particular, from reviewer 3 include:a) That it is not clear that NGL2 expression was thoroughly validated in the 2013 study since sense controls were not presented along with the ISH experiments that placed NGL2 in horizontal cells, and antibody specificity was not validated on retinal sections or whole mount retinas from NGL2-/- animals.b) That TdTomato expression should itself be a proxy of CRISPR knockout of NGL2 in the targeted cell, but additional analysis would help rule out the possibility that NGL2 has not been removed from neighboring bipolar cells (BPs), where it might lead to the phenotypes described here.c) Use of a BP cell marker (perhaps PKCalpha and/or mGluR6) for imaging (high resolution confocal imaging should be tried) with NGL2 and TdTomato here would provide additional support for your conclusions.d) Performing NGL2 staining of an AAV-YFP horizontal cell axon arbor control, and its quantification, to be able to compare it with the presented CRISPR-targeted NGL2 knockout cell in Figure 1.

Reviewer 3 raises concerns that the effects on horizontal cell axons and their synaptic connections with rod photoreceptors, which we observe following AAV-CRISPR deletion of NGL2 in the present study (Figure 2), and in NGL2^-/-^ mice in the present (Figure 5 and Figure 6) and a previous study (Soto et al., 2013), may be accounted for by effects on NGL2 in bipolar cells. The following arguments help dispel these concerns. 1) Our previous in situ hybridization and immunohistochemistry showed that in the outer parts of the inner nuclear layer, NGL2 is expressed exclusively in horizontal cells (Soto et al., 2013, Figure 1). 2) For our revisions, we injected retinas with either AAV-YFP to label horizontal cells or AAV-Grm6-YFP to label bipolar cells. In a new figure supplement (Figure 1—figure supplement 1), we show that NGL2 immunostaining in these retinas colocalizes with horizontal cell axon tips, but not with the tips of bipolar cell dendrites. 3) In our previous study (Soto et al., 2013, Figure 3), we did confirm the specificity of our NGL2 immunostaining. This specificity is also evident in Figure 5 (Figure 5—figure supplement 1) and Figure 6 (Figure 6—figure supplement 1) of our revised manuscript, in which we restore expression to individual horizontal cells in NGL2^-/-^ mice. NGL2 staining in these cases is restricted to single horizontal cell axon arbors. Together, these three points argue that NGL2 mRNA and protein are expressed in horizontal cells, but not in bipolar cells. 4) In our AAV-CRISPR strategy, tdTomato labels cells expressing sgRNAs. We observe effects of both sgRNAs in areas of the retina where individual horizontal cells and no bipolar cells are labeled with tdTomato. 5) In experiments, in which we restore expression of NGL2 to individual horizontal cells in NGL2^-/-^ mice, we find that axon size and synapse density are restored to and beyond wild-type levels, respectively. We observed these effects in areas in which no bipolar cells were infected with NGL2-expressing AAVs. These two points argue that horiztonal cell-specific gene deletion and gene delivery cause and correct, respectively, the observed phenotypes in horizontal cell morphology and connectivity.

4) Citations and relating your story to other published work: Tedeschi et al., Neuron, 2016, as mentioned by reviewer 2, describe that synapse formation and axon growth may be events that exclude one another, in a regeneration scenario, but nonetheless should be cited. There is also an interesting literature from the Hobert lab on C. elegans and axon/synapse maintenance, first published over 10 years ago (Aurelio et al., Science 2002), and recently by Y. Jin's lab (Neuron, 2016 PMID: 26796686, that might be relevant.

Following the reviewers’ suggestion, we discuss the relationship between synapse formation/stability and axon growth/maintenance in our revised manuscript, relating our findings to the previous literature including the studies mentioned by the reviewers.

Should you be able to acknowledge that the tips of the axons were not verified as synapses (and/or do the minimum to verify synapses), and sufficiently amend #2 on rescue and #3 on protein validation in two months, and the study is deemed appropriate for publication, then the hard-core validation of synapse structure using immune-EM could be submitted and hopefully published as an "Advance" (see https://elifesciences.org/articles/research-advance). The new eLife Research Advance mechanism is a short article that allows the authors of an eLife paper to publish new results that build on their original research paper in an important way.Reviewer #1:The manuscript by Soto et al. investigates the role of the synaptic cell adhesion molecule NGL2 in adult retinal horizontal cells (HCs). The authors use a clever CRISPR method to delete NGL2 from sparse HCs at various time points after early development. The data presented here show that NGL2 cell autonomously constrains HC axon growth while positively regulating synapse-forming structures in mature neurons. Unfortunately, I have two major concerns with the present study.1) The data are overall of good quality but the new knowledge gained from this study only incrementally advances results reported in their more extensive 2013 J. Neuroscience paper. Essentially, the new study shows that NGL2 has the same functions in young and old neurons. Advancing the field more substantially will require more experiments in a new direction, possibly investigation of NGL2 signaling or ligands/receptors during these processes.

To advance our knowledge of the molecular mechanisms that regulate circuit development and maintenance further, we generated new AAVs to express NGL2 in individual horizontal cells in NGL2^-/-^ and wild-type mice. AAV-mediated NGL2 expression exceeded wild-type levels. Restoring NGL2 to individual horizontal cells in NGL2^-/-^ mice restored axon size of infected cells to wild-type levels and increased synapse density beyond wild-type levels. This was true irrespective of whether AAVs were injected at P0 (i.e. before phenotype develops in NGL2^-/-^ mice) or at P30 (i.e. after phenotype develops in NGL2^-/-^ mice). Overexpression of NGL2 in individual horizontal cells in wild-type mice increased synapse density in infected compared to uninfected cells. This was observed both for injections at P0 and P30. These results, which we present in Figure 5 and Figure 6 of our revised manuscript, demonstrate that NGL2 can drive synapse formation and axon retraction in developing and mature horizontal cells. Together with our AAV-CRISPR experiments, AAV-mediated rescue and overexpression reveal that NGL2 levels restrict axon size and control synapse density in developing and mature retinal circuits in a bidirectional cell-autonomous manner.

2) The authors exclusively use axon tips as a proxy for synapses throughout this study but they never actually analyze bona fide synapses or synaptic markers other than NGL2. They often state "synapse tips" in the text and figure legends when "axon tips" should be stated. In their previous paper, they show that about 75% of axon tips contact rod photoreceptors. Here, it is not actually shown how HC-rod synapses may be affected by loss of NGL2 in adult neurons. How does adult loss of NGL2 affect axon tips contacting rods versus axon tips not contacting rods? One can imagine that existing HC-rod synaptic contacts may be more stable and harder to disperse than HC axon tips not in contact with rods. The main conclusions of the paper would be much stronger if true synapse contacts were examined after adult, conditional loss of NGL2, possibly using immuno-EM methods.

We thank the reviewer for raising this important point. We have not (nor has anyone else to our knowledge) previously shown that only about 75% of horizontal cell axon tips contact rod photoreceptors. For our revisions, we stained for the presynaptic ribbon anchoring protein Bassoon. We found that in control conditions (96.7% ± 1.3%, n = 7) and after removal of NGL2 (97.7% ± 1.5%, n = 4) nearly all horizontal cell axon tips were apposed by presynaptic ribbons. We conclude that axon tips can be used to count synapses between horizontal cells and rod photoreceptors, and that the reduced density of axon tips caused by removal of NGL2 reflects synapse loss. We present results from Bassoon staining in a new figure supplement (Figure 1—figure supplement 1) and explain our use of axon tips as indicators of synapses in the Results section of our revised manuscript.

Reviewer #2:[…] 1) Potential off-target effects: The sgRNA approach to ablate NGL2 in the Cas9 transgenic mice is very nice. However, the authors should try re-expressing NGL2 in the NGL2 ablated horizontal cells. Is the phenotype then reverted to wild type condition?

We generated new AAVs to express NGL2 in individual horizontal cells in NGL2^-/-^ and wild-type mice. AAVmediated NGL2 expression exceeded wild-type levels. Restoring NGL2 to individual horizontal cells in NGL2^-/-^ mice restored axon size of infected cells to wild-type levels and increased synapse density beyond wild-type levels. This was true irrespective of whether AAVs were injected at P0 (i.e. before phenotype develops in NGL2^-/-^ mice) or at P30 (i.e. after phenotype develops in NGL2^-/-^ mice). Overexpression of NGL2 in individual horizontal cells in wild-type mice increased synapse density in infected compared to uninfected cells. This was observed both for injections at P0 and P30. These results, which we present in Figure 5 and Figure 6 of our revised manuscript, not only support the specificity of our sgRNA approach, but also demonstrate that NGL2 can drive synapse formation and axon retraction in developing and mature horizontal cells. Together with our AAV-CRISPR experiments, AAV-mediated rescue and overexpression reveal that NGL2 levels limit axon size and control synapse density in developing and mature retinal circuits in a bidirectional cell-autonomous manner.

2) Synapses: If the reviewer understands the study correctly, the authors did not use synaptic markers but focus on axon tips when they present their results on synapse reduction in NGL2-ablated horizontal cells. Can the authors perform a co-staining with a presynaptic marker (synapsin, synaptphysin)?

Following the reviewer’s suggestion, we stained for the presynaptic ribbon anchoring protein Bassoon to test what fraction of horizontal cell axon tips are synaptic in control conditions and after removal of NGL2. Both in control conditions (96.7% ± 1.3%, n = 7) and after removal of NGL2 (97.7% ± 1.5%, n = 4) nearly all horizontal cell axon tips were apposed by presynaptic ribbons. We conclude that axon tips can be used to count synapses between horizontal cells and rod photoreceptors, and that the reduced density of axon tips caused by removal of NGL2 reflects synapse loss. We present results from Bassoon staining in a new figure supplement (Figure 1—figure supplement 1) and explain our use of axon tips as indicators of synapses in the Results section of our revised manuscript.

3) Statistics: Can the authors write a bit more detailed regarding the statistical tests they used? When they have multiple groups, do they use the Bonferoni corrections?

When comparing multiple groups (Figure 2, Figure 5, and 6), we used ANOVA testing with Bonferroni corrections. We include this information in the respective figure legends of our revised manuscript.

4) A recent paper (Tedeschi et al., Neuron, 2016) proposed that axon growth and synapse formation may be cellular events that exclude each other. The authors should discuss their findings in the context with this paper.

As suggested, in our revised manuscript, we discuss our findings in the context of other studies exploring the relationship of synapse formation/stability to axon growth/maintenance, including the work of Tedeschi et al. (2016).

Reviewer #3:[…] 1) The present paper depends critically upon light-level assessment of NGL-2 expression and axon tips from CRISPR-manipulated and non-CRISPR-manipulated GFP-expressing HCs, and assumptions about how this association defines synaptogenesis between HC axons and rod photoreceptors. However, the previous study by these investigators (Soto et al., 2013) upon which this present study is based leaves open a few questions that would be very helpful for the authors to address here in order to strengthen this study. First, is not clear that NGL2 expression was thoroughly validated in the 2013 study since sense controls were not presented along with the ISH experiments that placed NGL2 in horizontal cells, and antibody specificity was not validated on retinal sections or whole mount retinas from NGL2-/- animals. These two simple experiments should be done to support this present communication.Further, if the NGL2 protein signal clearly overlapped with TdTomato+ horizontal cell axon tips, it would lend strength to the argument that NGL2 is indeed expressed presynaptically at horizontal cell axon/rod photoreceptor synapses. However, the signal in panel D from Figure 2 of Soto et al., 2013 does not appear to overlap with the labeled horizontal cell. Rather, the signal appears juxtaposed with the TdTomato signal, calling into question whether or not it actually arises from co-labeling in HC axons. This NGL2 labeling pattern adjacent the axonal arbor of an otherwise intact, wild type, HC observed in Soto et al., 2013, Figure 2, is reminiscent of the signal presented in the current study for a proposed NGL2 CRISPR knockout horizontal cell (Figure 1), which weakens the argument that NGL2 protein is absent in the targeted horizontal cell because of the proposed CRISPR-mediated NGL2 excision event. Although TdTomato expression should itself be a proxy of CRISPR knockout of NGL2 in the targeted cell, additional would help rule out the possibility that NGL2 has not been removed from neighboring bipolar cells (BPs), where it might lead to the phenotypes described here. Though weak, in the 2013 paper there is some NGL2 in situ in the INL at P10 and increasingly so at P20 (adult in situs were not presented in Soto et al., 2013). Therefore, use of a BP cell marker (perhaps PKCalpha and/or mGluR6) for imaging (high resolution confocal imaging should be tried) with NGL2 and TdTomato here would provide support for the authors' conclusions. In addition, some explanation for how the non-overlapping but adjacent HC axon/axon tip labeling of TdTomato /NLG2 aligns with the authors' contention that these are both labeling the same presynaptic sites should be provided.

As requested, we have performed additional experiments to dispel concerns that our results could be due to manipulations of NGL2 in bipolar cells. Here, we summarize briefly the evidence (old and new) supporting that NGL2 regulates morphology and connectivity of horizontal cells in a cell-autonomous manner. This evidence falls into two categories: 1) NGL2 is expressed in horizontal cells but not in bipolar cells; 2) our manipulations of NGL2 target horizontal cells but not bipolar cells.

Re 1): Although we did not include sense controls in our previous study (Soto et al., 2013), in situ hybridization, which we combined with immunostaining for the horizontal cell-specific marker calbindin, suggested that NGL2 mRNA is expressed highly in horizontal cells. Based on its laminar position, the weaker fluorescent in situ signal in the inner nuclear layer overlapped with amacrine rather than bipolar cells (Soto et al., 2013, Figure 1). In our previous study, we did confirm the specificity of immunostaining for NGL2, showing that it is absent in NGL2^-/-^ mice (Soto et al., 2013, Figure 3). This specificity is also evident in in Figure 5 (Figure 5—figure supplement 1) and Figure 6 (Figure 6—figure supplement 1) of our revised manuscript, in which we restore expression to individual horizontal cells in NGL2^-/-^ mice. NGL2 staining in these cases is restricted to single horizontal cell axon arbors. As suggested by the reviewer, for our revision, we injected retinas with either AAV-YFP to label horizontal cells or AAV-Grm6-YFP to label bipolar cells. In a new figure supplement (Figure 1—figure supplement 1), we show that NGL2 immunostaining in these retinas colocalizes with horizontal cell axon tips, but not with the tips of bipolar cell dendrites. The observation that NGL2 staining does not overlap completely with the AAV- YFP signal, is likely accounted for by the fact that NGL2 is a transmembrane protein, whereas YFP is distributed in the cytosol. Together, our old and new data suggest the NGL2 mRNA and protein are expressed in horizontal cells, but not in bipolar cells.

Re 2): In our AAV-CRISPR strategy, tdTomato labels cells expressing sgRNAs. We observe effects of both sgRNAs used to delete NGL2 in areas of the retina where individual horizontal cells and no bipolar cells are labeled with tdTomato. For our revisions, we performed additional experiments, in which we restored expression of NGL2 to individual horizontal cells in NGL2^-/-^ mice. We find that this restores axon size and synapse density to and beyond wild-type levels, respectively. We observed these effect in areas in which no bipolar cells were infected with NGL2-expressing AAVs. Together, these experiments demonstrate that horiztonal cell-specific gene deletion and gene delivery cause and correct, respectively, the observed phenotypes in horizontal cell morphology and connectivity.

The evidence outlined in Re 1) and Re 2), in our opinion, convincingly shows that NGL2 regulates morphology and connectivity of horizontal cells in a cell-autonomous manner.

2) Related to the point raised above, it is problematic to refer categorically to "fewer axon tips," in the absence of a thorough characterization of synaptic structure and function (as was performed in Soto et al., 2013), as a proxy for "fewer synapses." References in the text to "fewer synapses" should be toned down to indicate fewer axon tips, or at least some qualification should be mentioned, unless of course additional synaptic markers are included in these analyses.

Following the reviewer’s suggestion, we stained for the presynaptic ribbon anchoring protein Bassoon to test what fraction of horizontal cell axon tips are synaptic in control conditions and after removal of NGL2. Both in control conditions (96.7% ± 1.3%, n = 7) and after removal of NGL2 (97.7% ± 1.5%, n = 4) nearly all horizontal cell axon tips were apposed by presynaptic ribbons. We conclude that axon tips can be used to count synapses between horizontal cells and rod photoreceptors, and that the reduced density of axon tips caused by removal of NGL2 reflects synapse loss. We present results from Bassoon staining in a new figure supplement (Figure 2—figure supplement 1).

3) It is critical to see NGL2 staining of an AAV-YFP horizontal cell axon arbor control, and its quantification, so as to be able to compare it with the presented CRISPR-targeted NGL2 knockout cell in Figure 1. In addition, one assumes the non-Cas9 WT young and mature adult mice show none of the phenotypes indicated here for the Cas9 adult mice.

As suggested, we have included data quantifying the intensity of NGL2 staining in AAV-YFP-expressing horizontal cell axons in Figure 1 of our revised manuscript.

In addition, we obtained data from mature adult WT mice injected with sgRNAs and in so doing confirmed that without Cas9 sgRNAs do not affect horizontal cell morphology, irrespective of age. We present these data in a new figure supplement (Figure 4—figure supplement 1) of our revised manuscript.

4) Figure 2, describing HC axon contacts with cones, appear not to be discussed in the body of the Results section. Please include referral to these data in the main text.

We thank the reviewer for pointing this oversight out to us. We have added a reference to Figure 2, which present data on contacts between horizontal cell dendrites and cones, to the Results section of our revised manuscript.

5) The conclusion that "mistargeting in the germline knockout involves non-cell-autonomous actions of NGL2" requires qualification. It seems quite plausible that HC axon growth leading to stray processes could result from incomplete knock down of NGL2, and indeed these data presented here showing reduced, but not completely absent, NGL-2 association with axon tips from sgRNA-expressing HCs support this idea.

It seems unlikely that incomplete knock down of NGL2 leads to stray processes of horizontal cell axons for the following reasons. First, stray processes are more frequent in NGL2^-/-^ mice, in which NGL2 is completely absent. Second, the reduced but not completely absent NGL2 labeling is explained by the observation that two or more horizontal cells innervate each rod (see Figure 1). In our revised manuscript, we have qualified the statement concerning mistargeting to include the possibility that delays in the removal of NGL2 in addition to non-cell-autonomous actions of NGL2 could account for the lower frequency of mistargeted axon processes in horizontal cells expressing sgRNAs compared to horizontal cells in NGL2^-/-^ mice.

[Editors' note: further revisions were requested prior to acceptance, as described below.]

[…] Nonetheless, one of the reviewers cited some remaining issues that need to be addressed before acceptance, as outlined below:1) Considering, and naming of, axon tips as synapses: You have nicely demonstrated that horizontal cell axon tips almost always end in a synapse as indicated by the presence of the presynaptic marker bassoon; this validates their use of axon tips as a proxy for synapses. However, in the remainder of the paper, you still count axon tips and not synapses; thus, it would be more accurate to refer to the axon tips as such and not as "synapses" or "synapse tips". This consideration is similar to counting spines. In the literature, one would not count spines for an experiment and then state that synapse density is changed. You would still state that spine density is changed, which likely reflects altered synapse number.

We have revised our manuscript as suggested by the reviewer. When describing our results, we now consistently refer to axon tips. Because we show that nearly all axon tips are synaptically differentiated, we interpret changes in the density of axon tips as evidence for changes in synapse density. We explain our use of axon tips as proxies for synaptic connectivity early in the Results section.

Also, the Materials and methods do not list the bassoon antibody, its source, etc., nor describe the details of the staining experiment; please amend.

We have added this information to the Materials and methods section of our revised manuscript.

2) NGL2 expression in WT: In the NGL2 expression experiments (Figure 5 and Figure 6), did the virally expressed NGL2 have an epitope tag? If not, how were the NGL2-expressing cells identified in WT mice? Staining with the NGL2 antibody in WT mice should reveal a sea of NGL2 puncta. How were the images in 5E and 6E obtained? If a subtraction erasing of signal using Amira software was used as alluded to in the Materials and methods, the original images should be shown in a supplement. It is still unclear how they identified the overexpressing neurons because the images in the supplements for Figure 5 and Figure 6 are insufficient to support the stated conclusion that "AAV-mediated expression of NGL2 exceeded WT protein levels". WT levels should be shown for comparison and if possible, some kind of quantification is warranted or the statement revised.

The virally expressed NGL2 did not have an epitope tag. We identified infected cells by staining with the NGL2 antibody. The intensity of NGL2 staining in infected cells was considerably higher than in wild-type cells. This can be seen in panels B of Figure 5—figure supplement 1 and Figure 6—figure supplement 1. These panels show original images adjusted to reveal wild-type labeling at the expense of saturating the labeling of infected horizontal cells. Conversely, panels E of Figure 5 and Figure 6, which show original images of the same cells, were adjusted not to saturate labeling of infected axons. As a consequence, the wild-type NGL2 labeling is not visible in these panels. In *Ngl2^-/-^* retinas, there is no NGL2 staining outside the infected axons (s. panels A of the figure supplements). We explain the differences between images in the main Figures and figure supplements in more detail in the legend of the figure supplements in our revised manuscript. The difference in staining intensity of AAV-mediated and wild-type NGL2 expression is, in our opinion, clear without quantification.

3) Statistics: The authors indicate that they conducted multiple comparisons after each ANOVA and p-values are often listed after particular testing conditions in the figure legends. However, identification of the groups the p-value describes is usually not present. For example, are the p values for the ANOVA or for some post-hoc multiple comparison? If a multiple comparison, which groups? It would be easier to read and quickly interpret if the p-values were indicated in the graphs themselves or in a separate chart as a supplement. This comment pertains to most of the figures. Finally, we recommend that you provide more information about the statistics as described in comment above.

We used ANOVA testing to compare data in Figure 2, Figure 5, and 6. As mentioned in the respective legends, the p-values for these figures are calculated with Bonferroni corrections for multiple comparisons. Following the reviewer’s suggestion, we indicate more clearly the groups that were compared in the legends of these figures in our revised manuscript.

4) Graph in Figure 1 has no x-axis label.

We have added the label “YFP | tdT” to the x-axis to indicate that the data plotted along the y-axis are from pairs of horizontal cells labeled with these fluorescent proteins.

5) Reference to Figure 2 is not referred in the text, nor any explanation of the experiment provided.

Thank you for pointing this oversight out to us. We have added a reference to these figure panels and mention the experiments they represent in the Results section of our revised manuscript.